# An Overview on Diffuse Large B-Cell Lymphoma Models: Towards a Functional Genomics Approach

**DOI:** 10.3390/cancers13122893

**Published:** 2021-06-09

**Authors:** Natalia Yanguas-Casás, Lucía Pedrosa, Ismael Fernández-Miranda, Margarita Sánchez-Beato

**Affiliations:** 1Lymphoma Research Group, Medical Oncology Department, Instituto de Investigación Sanitaria Puerta de Hierro-Segovia de Arana (IDIPHISA), C/Joaquín Rodrigo 2, Majadahonda, 28222 Madrid, Spain; lpedrosa@idiphim.org (L.P.); ifernandez@idiphim.org (I.F.-M.); 2PhD Programme in Molecular Biosciences, Doctorate School, Universidad Autónoma de Madrid, 28049 Madrid, Spain; 3Centro de Investigación Biomédica en Red de Cáncer (CIBERONC), 28029 Madrid, Spain

**Keywords:** B-cell lymphoma, diffuse large B-cell lymphoma, genomics, functional model, cell lines, murine models, gene editing, organoid/spheroid, therapy, cancer

## Abstract

**Simple Summary:**

Lymphoma research is a paradigm of integrating basic and applied research within the fields of molecular marker-based diagnosis and therapy. In recent years, major advances in next-generation sequencing have substantially improved the understanding of the genomics underlying diffuse large B-cell lymphoma (DLBCL), the most frequent type of B-cell lymphoma. This review addresses the various approaches that have helped unveil the biology and intricate alterations in this pathology, from cell lines to more sophisticated last-generation experimental models, such as organoids. We also provide an overview of the most recent findings in the field, their potential relevance for designing targeted therapies and the corresponding applicability to personalized medicine.

**Abstract:**

Lymphoma research is a paradigm of the integration of basic and clinical research within the fields of diagnosis and therapy. Clinical, phenotypic, and genetic data are currently used to predict which patients could benefit from standard treatment. However, alternative therapies for patients at higher risk from refractoriness or relapse are usually empirically proposed, based on trial and error, without considering the genetic complexity of aggressive B-cell lymphomas. This is primarily due to the intricate mosaic of genetic and epigenetic alterations in lymphomas, which are an obstacle to the prediction of which drug will work for any given patient. Matching a patient’s genes to drug sensitivity by directly testing live tissues comprises the “precision medicine” concept. However, in the case of lymphomas, this concept should be expanded beyond genomics, eventually providing better treatment options for patients in need of alternative therapeutic approaches. We provide an overview of the most recent findings in diffuse large B-cell lymphomas genomics, from the classic functional models used to study tumor biology and the response to experimental treatments using cell lines and mouse models, to the most recent approaches with spheroid/organoid models. We also discuss their potential relevance and applicability to daily clinical practice.

## 1. Unraveling Aggressive B-Cell Lymphomas: The Elusive Link between Genomics and Personalized Treatment

Lymphoma research is a paradigm of the integration of basic and applied research within the fields of molecular-marker-based diagnosis and therapy. Basic research on lymphomas has deepened our knowledge of cancer mechanisms, while helping to elucidate how aging, infection, the immune system, and other genetic and environmental factors might account for the increasing incidence of this disease in western countries. The rapid rise of high-throughput technologies in recent years has led to a growing number of genomic studies that emphasize and characterize the high degree of genomic heterogeneity in lymphomas. Genetic classification and other phenotypic and clinical data are currently used to predict which patients could benefit from standard treatment. However, new therapies that could be of use to patients at higher risk from refractoriness or relapse have yet to be applied in daily clinical practice. Indeed, the proposal of alternative treatments for diffuse large B-cell lymphoma (DLBCL) patients who do not respond to standard immunotherapeutic regimens (up to 30–40%) is mostly empirical, based on trial and error, and does not consider their genetic complexity. This is primarily due to the intricate mosaic of genetic and epigenetic alterations in lymphomas, and the unpredictable effects on their phenotype, which make it difficult to foresee which drug will work for any given patient. Epigenetic processes, such as methylation and demethylation, are responsible for the dynamic plasticity of B-cells and affect tumor behavior (for a recent review, see [1]). Interestingly, many of the genetic mutations that occur in DLBCL affect genes involved in chromatin remodeling (*EZH2*, *CREBBP*, *KMT2D*, etc.) and other epigenetics functions such as DNA methylation (*TET2*).

Patient treatment based on genomic analysis alone is not straightforward. Matching a patient’s genes or a specific genetic signature to drug sensitivity by directly testing live tissues is the basis of the concept of precision medicine. However, in the case of lymphomas, we need to expand this concept beyond genomics to eventually provide better treatment options for patients in need of alternative therapeutic approaches. Functional assays could help fill this gap. Established models, such as those of tumor cell lines and mice, do not completely capture the complex genetic and epigenetic landscape that each patient presents, and, therefore, the translation of findings from these models to the clinic remains inefficient. Three-dimensional (3D) models, such as organoids, which can be generated from several cell types and reproduce the interactions of a patient’s immune system with their tumor, will be a promising tool in personalized medicine in the near future.

## 2. Diffuse Large B-Cell Lymphomas: Clinically Unresolved Genomic Complexity

### 2.1. Diffuse Large B-Cell Lymphoma: A Traditional Overview

DLBCL is an aggressive and heterogeneous disease with a variable clinical outcome. It is the most common non-Hodgkin lymphoma (NHL) subtype, with 3.13 and 5.6 new cases diagnosed per 100,000 habitants per year in Europe [2] and the United States [3], respectively. DLBCL is subdivided into morphological variants, molecular subtypes and different clinical entities (for a recent review, see [4]). The cases classified as DLBCL not otherwise specified (DLBCL-NOS) are biologically heterogeneous and the current classification for this lymphoma is based on its cell of origin (COO). Two subtypes are recognized based on the gene expression profile: activated B-cell (ABC) and germinal center B-cell (GCB) DLBCL [5,6]. Around 10–15% of cases cannot be classified as either of these subtypes and, therefore, remain unclassified. This classification is of prognostic value, with ABC-DLBCL being associated with poorer clinical outcome.

The most recent revision of lymphoid neoplasms by the WHO [7,8] recognized a more aggressive, high-grade B-cell lymphoma subtype with a worse prognosis, the ‘double-hit’/‘triple-hit’ (DH/TH) lymphoma, which is characterized by *MYC* rearrangements that are associated with *BCL2* and/or *BCL6* rearrangements. Lymphomas that co-express the MYC and BCL2 proteins, known as double-expressor lymphomas, are relatively frequent, and have a worse prognosis than other DLBCL-NOS, although their behavior is not as aggressive as that of the DH/TH lymphomas [9].

In recent years, several studies using massive sequencing analysis have helped to define the genetic landscape of DLBCL by identifying many genetic alterations. Primarily, DLBCL has alterations in genes involved in B-cell differentiation pathways, such as chromosomal rearrangements of the *BCL6* locus and mutations/deletions in *PRDM1* [10,11,12] and *IRF8* [13,14] and BCR/NF-κB signaling, such as *CD79B*/*A*, *BCL10*, *TNFAIP3*, and *CARD11* [11,12,14,15,16,17,18,19,20]. Other relevant pathways are also affected, such as the Toll-like receptor pathway, with mutations in *MYD88* and, less frequently, in *TRAF6* [11,12,14,17,18,19,20,21]; the p53 and DNA damage pathway, with deletions/mutations in *TP53* and *UBE2A* [11,14,17,19,22], and methylation/deletions of *CDKN2A* [23]; and the PI3K/AKT pathway with mutations in *PTEN* and *PIK3CA* [24]. Other genes that are frequently or significantly altered are *MYC* (which modulates many cellular functions, such as DNA replication and cell proliferation)*, BCL2* (cell apoptosis), *NOTCH1*, *NOTCH2,* and *DTX1* (cell differentiation), *GNA13* (cell migration), *CD58* and *B2M* (immune evasion), *PIM1*, *BTG1,* and *CCND3* (cell cycle), and the epigenetic regulators *EZH2*, *HIST1H1E*, *HIST1H1C*, *KMT2D*, *CREBBP,* and *EP300* [11,12,14,17,18,19,20,25,26,27,28,29,30] (Figure 1).

For some genes, the mutation rate differs depending on the COO subtype. ABC-DLBCLs have a high frequency of alterations in *MYD88* (MYD88^L265P^ occurs almost exclusively in ABC-DLBCL) *PRDM1*, *CD79B*, *BCL10*, and *BCL6* rearrangements [26,28,30]. GCB-DLBCLs have a better prognosis and frequently feature mutations in *BCL2, GNA13, TNFSR14*, *EZH2*, *CREBBP,* and *BCL2* rearrangements [26,28,30].

Around 30% of DLBCL cases show rearrangements of the 3q27 region, involving *BCL6*, most commonly in the ABC subtype. *BCL2* rearrangements (t(14;18), a hallmark of follicular lymphoma), have been detected in 20–30% of DLBCL cases, usually in those of GCB, and the *MYC* rearrangement is observed in 8–15% of cases [8]. When *MYC* translocation occurs simultaneously with *BCL2* and/or *BCL6*, the cases are classified as DH/TH high-grade B-cell lymphoma, as mentioned previously [28,30].

### 2.2. Advances in DLBCL Genomics: Genetic Signature Helps Design Tailored Therapies

In recent years, new DLBCL genetic classifications have emerged based on specific genetic alterations defining different subtypes. In 2017, Reddy et al. presented a multivariate model using gene expression data and COO classification in combination with genetic alterations to classify patients into three risk groups with different survival outcomes [29]. A year later, Schmitz et al. [28] identified four genetic profiles—MCD, BN2, N1, and EZB—based on mutational and translocation data (Table 1). These were subsequently refined by the addition of two subtypes: ST2 and A53 [31]. MCD was characterized by mutations in *MYD88*^L265P^ and *CD79B*, while N1 presented mutations in *NOTCH1*. In both subtypes, most cases were ABC-DLBCL, and had a worse prognosis than BN2, based on *BCL6* fusions and *NOTCH2* mutations, and EZB, which was characterized by *BCL2* translocations and mutations in the *EZH2* gene [28]. The ST2 subtype was characterized by recurrent mutations in *SGK1* and *TET2.* Most cases were GCB-DLBCL and had a better survival. The A53 subtype comprised aneuploidy cases with mutations and deletions in *TP53*, with a worse prognosis among ABC-DLBCLs [31]. Another important study, by Chapuy et al. [32], defined five subtypes, including two ABC-DLBCL groups, one with lower risk and a possible marginal zone origin (C1, with *BCL6* translocations and mutations in *NOTCH2*), and the other a high-risk group (C5) featuring a high frequency of cases with mutations in *MYD88*, *CD79B*, and *PIM1*; they also described two distinct subsets of GCB-DLBCLs with favorable (C4) and poor outcomes (C3), and an ABC/GCB-independent group (C2) with biallelic inactivation of *TP53*, *CDKN2A* loss, and associated genomic instability [32]. Similar results from a large real-world population-based patient cohort in which MYD88 (similar to MCD) and NOTCH2 (similar to BN2) subtypes had a worse prognosis than BCL2 (EZB), SOCS1/SGK1, and TET2/SGK1 subtypes, which, together, could be considered similar to ST2 [33] (Table 1). More recently, another study by Pedrosa et al. [34] classified samples in MCD^2-S^ (*MYD88*, *CD79B*, *PIM1*, *PIM2*, *PRDM1*, *BTG1,* and *CD58*), BN2^2-S^ (*BCL6* fusion, *NOTCH2*, *BCL10*, *TNFAIP3*, *UBE2A*, *CD70*, *CCND3,* and *DTX1*), N1^2-S^ (*NOTCH1*), EZB^2-S^ (*BCL2* fusion and mutation, *EZH2*, *CREBBP*, *TNFRSF14*, *KMT2D IRF8,* and *EP300*), and ST2^2-S^ (*SOCS1*, *SGK1*, *TET2,* and *STAT3*) with a two-step classifier based on the studies by Schmitz et al., Chapuy et al., and Lacy et al. [28,32,33]. Furthermore, the Lacy cohort showed a strong consensus across three classification strategies [34,35], reinforcing the high degree of reproducibility of these subtypes.

Knowledge of these alterations may allow for the selection of better treatments than the standard rituximab plus cyclophosphamide, doxorubicin, vincristine, and prednisone (R-CHOP), based on the patient’s genetic background. Furthermore, it may prompt the development of novel agents or strategies targeting selected combinations of genes and pathways that synergistically affect the currently available drugs. These target survival pathways are driven by specific genetic and epigenetic events, such as BCR signaling, of which BTK, PI3K, and mTOR inhibitors stand out, and other agents targeting epigenetic modifiers, apoptosis, and the immune system regulator or the NF-κB pathway. To date, despite the arrival of new agents in the last decade, the actual treatment of DLBCL has changed little since the introduction of immunochemotherapy [26,36,37], and further effort is needed if precision medicine is to be successfully applied in the treatment of this pathology.

### 2.3. Friend or Foe: The DLBCL Microenvironment

Despite the intrinsic relevance of genetic aberrations within lymphoma cells, DLBCL complexity comprises more than genetic alterations in the tumoral cells. DLBCL cell survival and physiology depend on their interactions with the non-malignant cells and stromal elements, which constitute the tumor microenvironment (TME). The DLBCL TME involves the interactions of lymphoma cells with host immune cells, including natural killer (NK) cells (±20% of total cell content), dendritic cells (DCs) (±15%), M2-type macrophages (±15%), CD4+ T cells (±10%), and CD8+ T cells (<5%) [38]. Significantly, TME composition and its highly heterogeneous interactions with neoplastic cells determine DLBCL pathogenesis and progression [39].

Tumor cells account for 60–80% of the cell content in DLBCL. The acquired genetic mutations render tumor cells relatively independent of survival and proliferation signals from their microenvironment [40]. Regardless of this effaced TME composition pattern, it has an important impact on DLBCL patients’ survival, therapy response and disease progression or relapse. Interestingly, a recent work has described four major DLBCL microenvironment categories associated with distinct biological aberrations and clinical behavior [41]. Indeed, TME is of predictive importance in DLBCL prognosis and has been linked to resistance to chemotherapy [24,38,42]. This prompted the development of drugs targeting TME composition and the blockade of cellular interactions with dissimilar results in clinical trials. Therapeutic blockade of CD47, in combination with rituximab, showed promising response rates in DLBCL patients, while the PD-1 blockade response fell short (for a review, see [43]). On the other hand, the immune modulator lenalidomide has shown promising results in monotherapy and in combination with R-CHOP regimens, especially in elderly ABC-DLBCL patients [40].

In conclusion, lymphomas arise within a complex microenvironment and are characterized by a considerable genetic heterogeneity. Therefore, the ideal DLBCL model should replicate both aspects, and the clinical heterogeneity of the pathology.

## 3. Functional Models for B-Cell Lymphoma Research

From its outset, lymphoma research has relied on a wide variety of models, from cell monolayers to whole-organism studies (predominantly mice). Over the years, advances in gene-editing techniques, mouse models, and the genesis of three-dimensional (3D) in vitro models have led to new approaches to the study of DLBCL biology and therapy responses in a more physiological environment. The generation of reliable models approximating human disease offers the potential to apply the obtained results in daily clinical practice.

### 3.1. Gene Editing of Lymphoma Cells

The current gene-editing techniques include DNA base editors and RNA-programmable clustered regularly interspaced short palindromic repeats (CRISPR)-associated (Cas) nucleases [44,45]. Gene editing with these systems is effective in cell lines, human samples, and murine models [46,47,48]. Both strategies allow for the generation of accurate lymphoma models that reproduce or revert the mutations identified by massive sequencing, to study their relevance to tumor behavior and drug response [46,47]. Moreover, synthetically engineered DLBCL models, generated immediately by the directed mutations in human cell lines, or by their combination with mice of a specific genetic background, have enabled co-operating genetic alterations to be studied [46,48], such as FOXP1 silencing, BCR pathway expression or silencing, the co-expression of BCL2/MYC or BCL2/BCL6, and the expression of GC-DLBCL or ABC-DLBCL-related genes [29,49,50].

Beyond the insertion of specific mutations in a certain region of interest, the CRISPR-Cas9 system was adapted for functional genomic screening in DLBCL cell lines. Unbiased high-throughput CRISPR-Cas9 knockout screenings of DLBCL cell lines prompted the study of the functional impact of potential genetic drivers of DLBCL, previously identified by integrative analysis of whole-exome and transcriptome sequencing [29,51].

Therefore, gene editing is a useful tool for investigating genetic and functional drivers, and tumor dynamics, in DLBCL. However, there are hurdles to overcome before gene editing can be applied in a therapeutic context, such as ensuring a higher base-editing efficiency with minimal off-target effects, establishing a way of delivering the gene-editing system in a target organ with a high degree of correction, and it being possible to edit 100% of the tumor cells. Cancer is not a monogenic disease, and multiple somatic genetic alterations are involved. However, CRISPR-Cas9 has shown promising results in preclinical studies involving immunotherapy with CAR T-cells, indicating its therapeutic potential [52].

### 3.2. The Rise of Drug Screening: Mission Pathways

Cancer research has largely involved cell lines and 2D cell culture (Figure 2A), as they provide a robust and reproducible model system [53,54,55]. The universalization of massive parallel sequencing has facilitated the analysis of a large number of cancer model collections, such as the Cancer Cell Line Encyclopedia (CCLE), which includes some hematological cancer-derived cell lines [56]. However, limitations of the coverage of the hematological oncology spectrum in general collections prompted the development of the LL-100 panel, which covers more than 20 human leukemia and lymphoma entities, including DLBCL (ABC- and GC-DLBCL) [57]. The association of the knowledge emerging from the cell-line panels and the genetic studies of patient-derived lymphoma samples [31] has enabled the identification of potential therapeutic targets.

B-cell-lymphoma-derived cell lines have been widely used in the study of antitumor drugs. On the one hand, some strategies seek the functional confirmation of drug activity in these cell lines through the integration of omics-analysis with the use of isoform-specific inhibitors and/or small hairpin RNA (shRNA) or CRISPR-Cas9-mediated knockout [58,59,60,61]. On the other hand, other drug-screening approaches target lymphoma survival signaling pathways and critical alterations in DLBCL, such as those described below.

The B-cell receptor (BCR) pathway, key to B-cell survival, is an especially relevant therapeutic target, as its expression is restricted to B-cells, revealing potential pharmacological agents specific to these cells. Importantly, BCR signaling is chronically active in ABC-DLBCLs, with the consequent canonical NF-κB pathway activation [5,6,62]. Ibrutinib, a Bruton’s tyrosine kinase (BTK) inhibitor that blocks BCR-dependent NF-κB activation, is one of the candidates for tailored medicine for DLBCL patients, due to its strong effects in DLBCL-derived cell lines. In vitro approaches revealed a selective toxicity in ABC-DLBCL cell lines with mutations in *CD79A/CD79B* and MYD88^L265^ [15,63], suggesting a potential targeted therapy for ABC-DLBCL or the newly proposed genetic subtype MCD-DLBCL. However, clinical trials have shown that BTK inhibitors have a limited effect on DLBCL compared to other lymphoma subtypes [64].

NF-κB plays an important role in B-cell development. The treatment of DLBCL-derived cell lines with inhibitors of the multi-subunit IκB kinase (IKK) complex, the main activator of the NF-κB pathway, demonstrated that these drugs block the proliferation of ABC-DLBCL cells before inducing cell death [65]. In this way, in vitro approaches targeting the MYD88^L265^-promoted myddosome assembly, which results in a constitutive NF-κB activation, suggested that disruption of this structure inhibits cell survival and promotes apoptosis in ABC DLBCL with *MYD88*-mutations [66]. Lenalidomide, a thalidomide-derived immunomodulatory drug, also downregulates NF-κB activity, as it targets the E3 ubiquitin ligase component cereblon (CRBN) and downregulates IRF4. The effects of lenalidomide in ABC-DLBCL are associated with increased interferon β (IFN-β) production, which is pro-apoptotic, counteracting the effect of oncogenic *MYD88* mutations [61,67]. Therefore, MCD-DLCBL patients could benefit from lenalidomide therapy. Despite the increasing efforts in the discovery and generation of drugs targeting NF-κB, the clinical trials performed to date have limited applicability for DLBCL patients [68,69].

The JAK/STAT3 signaling pathway is also a key candidate for drug targeting. STAT3 and JAK1 are activated in ABC-DLBCL by autocrine production of the cytokines IL-6 and IL-10, promoting cell survival. The decrease in STAT3 activation induces apoptosis in DLBCL cell lines [70]. Additionally, JAK inhibitors decrease cell viability in ABC-DLBCL cell lines [59]. Both therapeutic strategies are promising candidates for ST2-DLBCL genetic subtype patients.

The PI3K/AKT pathway is another potential therapeutic target in B-cell lymphomas. There are several PI3K inhibitors, both pan-PI3K inhibitors and specific inhibitors of any of the PI3K isoforms. Their activity has been tested in several DLBCL-derived cell lines, and the results suggest that the simultaneous blocking of PI3Kα/δ is the most effective strategy in DLBCL treatment (47). Moreover, drug synergism studies proposed a combination of PI3Kα/δ and BCL2 blockade, using venetoclax, in BCR-dependent DLBCL cell lines [71], or even the use of a combination of PI3K and BTK inhibitors, to overcome ibrutinib resistance in ABC-DLCBL cell lines [72].

The therapeutic targeting of epigenetic regulators has also been studied in lymphoma-derived cell lines. One of the most important targets in this area is EZH2. The EZH2 inhibitor EZH2i blocks proliferation in *EZH2* mutant DLBCL cell lines [73], and even in *EZH2* wild-type lymphoma cells [29,31,74]. Moreover, dual targeting of EZH1/2 has been shown to be even more effective [75]. The results from these studies could form the basis for personalized medicine for EZB-DLBCL patients or GCB-DLBCL overexpressing *EZH2* wild-type and/or mutants [76]. Several studies of B-cell lymphoma cell lines showed that HDAC inhibitors can suppress tumor growth via multiple mechanisms, including the induction of cell death, up-regulation of MHC [77] and/or PD-L1 [78], and activation of Treg (for a recent review, see [79]). Interestingly, some HDAC inhibitors, such as romidepsin, panobinostat, vorinostat, and belinostat, are already approved and used for hematological cancers in daily clinical practice [79].

Finally, alterations in NOTCH signaling are also involved in lymphomagenesis, and it could therefore be a promising target for DLBCL therapy. There have been few studies in this field, although it has been shown that gamma-secretase inhibitors may be of therapeutic value in DLBCL [80], especially for N1 DLCBL patients. The therapeutic strategies for targeting this pathway remain largely unexplored.

Constitutive overexpression of BCL2 and MYC usually occurs in DLBCL, making them potential therapeutic targets. MYC is epigenetically modulated by the bromodomain and extra terminal (BET) protein. In vitro experiments provide a rationale for using BET inhibitors alone or in combination with BCL2 inhibitors for DHL/THL DLBCLs [81,82]. Additionally, BCL6 is overexpressed in over half of DLBCL cases, regardless the COO. Both BCL6 peptide inhibitors and small molecule inhibitors have been shown to have potent efficacy against B-cell lymphoma cells [83,84] and BCL6-dependent DLBCL cell lines [85]. These results are suggestive of a potentially good therapeutic strategy for BN2-DLBCL patients.

Unlike cell suspensions which are obtained directly from fresh biopsies, human cell lines are a self-renewing resource containing most genetic alterations in the original tumor [86]. The extensive propagation of cell lines and potential handling-derived stresses lead to genetic inconsistencies in cancer cell lines that cause apparently standardized cells to behave differently from one another [87]. Additionally, the lack of microenvironment and other cell types in the culture, which are crucial in B-cell lymphomas, give rise to different transcriptomic and proteomic signatures in cell lines than in primary lymphomas [86].

### 3.3. Of Mice and Men: Understanding Lymphoma Biology

In the field of lymphoma research, the mouse has been the model organism due to its genetic and physiological similarity to humans (Table 2) [88,89]. Various spontaneous tumor models (such as the Eμ-Myc, Eμ-BRD2, or Bcl6 mouse models) have been developed to study how B-cell lymphomas arise and mature in different tumor environments [90,91,92]. However, genetically engineered mouse (GEM) models (Figure 2C) have enabled the recapitulation of DLBCL genomic complexity, and have been very useful for defining the genetic causes of lymphoma [93]. Indeed, these models have clarified the understanding of the role of BCR signaling, germinal center differentiation status (by the identification of factors such as BCL6, SPIB, PAX5, and EZH2, among others), B-cell development regulators (such as PI3K) in lymphomagenesis, the study of malignant transformation drivers (such as MEF2B) and tumor suppressors (such as TET2 or CREBBP), functional cooperation between B-cell pathways (such as BCR and TLR signaling), and the identification of potential targets for drug intervention [58,94,95,96,97,98].

The Eμ-Myc mouse model has been paradigmatic in our understanding of B-cell lymphomagenesis and the effect of novel therapies in lymphoma progression [90]. These mice carry a DNA construct of the murine c-Myc gene, in combination with an IgH enhancer, which leads to 90% of the heterogenetic offspring developing an early-onset immature form resembling BL, and a more indolent mature form resembling DLBCL [99]. This model has enabled the identification of essential factors for tumor survival (such as spleen tyrosine kinase (SYK) in NHL) and co-operative tumor suppressor genes in B-cell lymphoma. It has also led to the generation of new preclinical mouse models by the translocation of the N-myc gene in different enhancer regions combined with retroviral infection to accelerate and enhance lymphomagenesis, such as the Eμ-BRD2 DLBCL mouse model, or others that recapitulate complex lymphoma scenarios, such as aggressive B-cell lymphoma in the context of chronic lymphocytic leukemia [91,100,101,102].

**Table 2 cancers-13-02893-t002:** Currently available DLBCL mouse models (overview). DLBCL: Diffuse large B-cell lymphoma; GC: germinal center B-cell like; ABC: activated B-cell like; (m): murine origin; (h): human origin; iv: intravenous; ip: intraperitoneal; sc: subcutaneous; PDX: patient derived xenograft; PDO: patient derived organoid.

	Strategy	Phenotype/Incidence	Prospective Uses
**Genetically engineered mice**	Eμ-Myc	DLBCL (time dependent) [99]	
	Eμ-BRD2	DLBCL [91]	
	Bcl6 Knock in	GC-DLBCL [92]	
	Bcl6/Myc	ABC-DLBCL [92]	
	Iµ:HA.BCL6	36–62% lymphoma incidence [92]	Combination with conditional *Spen* and *Tnfaip3* knockout or oncogenic *Notch2* alleles to model Cluster BN2
	*Mb1:Cre;Eµ:Bcl2;Crebbp^fl/fl^*	GC-DLBCL [103]	Combination of the different alleles to generate a sophisticated EZB mouse model
	*Cγ1^Cre/wt^;Kmt2d^fl/fl^;VavP:Bcl2*	21% incidence GC-DLBCL [104]	
	Ezh2^cond.p.Y641F/wt^;VavP:Bcl2; Cγ1^Cre/wt^	DLBCL-like lymphoma [105]	
	Cd19^Cre/wt^;Myd88^cond.p.L252P/wt^;Rosa26^LSL.BCL2-IRES-GFP/wt^	85% incidence ABC-DLBCL [106]	Modeling of the MCD cluster by combination of both alleles and a newly generated *CD79B*^cond.p.Y196H/wt^ allele, or with the already existing *Prdm1^fl/fl^*
	*Cγ1^Cre^;Prdm1^fl/fl^;Rosa26^LSL.IKK2ca^*	50% incidence IRF4, post-GC DLBCL [107]	
**Syngeneic models**	Pi-BCL1 (m) iv or ip in BALB/c immunocompetent mice	DLBCL [108,109]	Gene editing Generation of complex organoids prior to inoculation of the cell line
	A20 (m) iv, intrasplenic, or sc injection in BALB/c immunocompetent mice	DLBCL [110,111,112]	
**Xenograft models**	SU-DHL-4 (h) iv or sc in SCID immunodeficient mice	DLBCL [113,114]	Gene editing Generation of organoids Inoculation in humanized NOD/SCID mice
	Transduced HPCs in irradiated mice	GC- and ABC-DLBCL [103,115]	Generation of a-la-carte HPCs reproducing the genetic signatures Inoculation in humanized NOD/SCID mice
	PDX iv or sc in immunodeficient mice	20–30% successful engraftment	Use of humanized mice to study lymphoma physiology and drug responses Personalized medicine
	PDO iv or sc in immunodeficient mice	20–30% successful engraftment	Genetic modification Use of humanized mice to study lymphoma physiology and drug responses Personalized medicine.

The headings of each mouse model subtype are highlighted in bold.

Targeted gene editing of mice initially generated germinal center (*Bcl6* knock-in) and post-germinal center (*Bcl6/Myc* transgenic) DLBCL mouse models [92]. In recent years, there has been a particular focus on the generation of models which capture the genomic complexity of the molecularly-defined DLBCL subtypes described in the Advances in DLBCL genomics section, by two distinct strategies: the generation of autochthonous mice, combining the use of conditional alleles with a certain genetic background, or by transplantation of hematopoietic progenitor cells (HPCs) transduced with selected repressors or enhancers in irradiated mice.

These strategies have given rise to mouse models with different genetic backgrounds, which can be used to study the recently described BN2, EZB, and MCD DLBCL subtypes (for a review, see [116]).

DLBCL samples belonging to the BN2 subtype are characterized by the co-occurrence of *BCL6* rearrangements with aberrations in *NOTCH2, SPEN*, *BCL10,* and *TNFAIP3*. The combination of conditional *Spen* and *Tnfaip3* knockout or oncogenic *Notch2* alleles (which are not sufficient to generate lymphomagenesis per se) with the *I**µ**.HA.BCL6* knock-in DLBCL mouse model, characterized by a deregulated BCL6 expression that leads to the development of human DLBCL-like lymphomas, might fit as a robust mouse model for this subtype [92,117,118,119]. However, no research has been carried out using this model to date.

The hallmark features of DLBCL samples belonging to the EZB subtype are genetic alterations in *BCL2* that lead to enhanced BCL2 expression, and mutations in genes encoding epigenetic modifiers such as *KMT2D, CREBBP,* or *EZH2*. The conditional deletion of *Kmt2d* mice early in the B-cell development process, combined with *Bcl*2 overexpression using *C**γ**1^Cre^;Kmt2d^fl/fl^;VavP:Bcl2* mice resulted in DLBCL lymphomagenesis [104]. The transplantation of *Kmt2d*-depleted HPCs into irradiated wild-type recipient mice resulted in clonal high-grade FL, not DLBCL, highlighting the functional relevance of the co-clustering of *BCL2* and *KMT2D* aberrations in these DLBCLs [120]. Likewise, *CREBBP* loss also cooperates with *BCL2* overexpression in DLBCL lymphomagenesis as shown in *Mb1:Cre;E**µ**:Bcl2;Crebbp^fl/fl^* mice and in an HPC transplantation model involving VavP:Bcl2-derived HPCs transduced with *Crebbp-* or *Ep300-*shRNA transplantation into irradiated mice [103,115]. Consistent with this, oncogenic mutations in *Ezh2,* combined with Bcl2 overexpression, also gave rise to early onset DLBCL lymphomas in *C**γ**1^Cre^;VavP:Bcl2;Ezh2^Y641F/N^* and in irradiated wild-type mice transplanted with *Ezh2^Y641F^*-transduced *VavP:Bcl2*-derived HPCs [105,121].

Co-occurring mutations in *MYD88* and *CD79B*, and *PRMD1* inactivation characterize MCD subtype DLBCLs. The B-cell-specific depletion of *Prdm1* leads to blocked plasma cell differentiation in mice (such as *Cd19^Cre^;Prdm1^fl/fl^* and *C**γ**1^Cre^;Prdm1^fl/fl^* mouse models). This has been combined with a constitutive active IKK2-bearing allele to generate ABC-DLBCL-bearing mice (*C**γ**1^Cre^;Prdm1^fl/fl^;Rosa26^LSL.IKK2ca^*). However, *IKK2* mutations are not recurrent in human DLBCL, making this mouse model a controversial one. Instead, the *Cd19^Cre/wt^;Myd88^p.L252P/wt^;Rosa26^LSL.BCL2-IRES-GFP^* mouse model contains the recurrent *Myd88^p^*^.L252P^ mutation and *BCL2* amplification in B-cells [106]. These mice develop lymphomas with late or post-GC origin ABC-DLBCL phenotype, with an actionable dependence on BCL2 [122]. This model can be further combined with the aforementioned *Prdm1^fl/fl^* allele to better resemble the genotype in this subtype. It would be ideal to generate a *CD79B* allele bearing the p.Y196H mutation (which disrupts the ITAM motif) and combine it with the aforementioned mouse models carrying the *Myd88^p^*^.L252P^ mutation for a better understanding of the lymphoma physiology of this subtype, as these mutations co-cluster in patients.

In short, this field is still laying the foundation to generate the perfect mouse model which reproduces the desired genomic complexity that will lead to genotype-tailored molecular intervention strategies for DLBCL patients. The application of the CRISPR-Cas9 system in GEM is a promising field of biomedical research as it can be performed directly, in a single step, in mouse zygotes by microinjection or electroporation, and allows for the simultaneous manipulation of multiple genes to mimic human disease complexity [123,124].

As useful as they are as a tool to understand the importance of certain genes or gene combinations in lymphomagenesis, GEMs lack the genomic complexity of the tumor, as they only reproduce selected alterations in a complex pathology.

Disseminated or widely distributed diseases (such as DLBCL) are efficiently reproduced by the intraperitoneal or intravascular injection of lymphoma cells via the tail vein in mice [109,125]. However, transplantable models of lymphoma involving the implantation of histocompatible allografts (syngeneic) or foreign xenografts into mice, maintain better tumor complexity, especially that due to the microenvironment.

DLBCL syngeneic mouse models are generated by the injection of A20 or Pi-BCL1 murine lymphoma cell lines into immunocompetent hosts. These models facilitate the study of lymphoma biology, tumor microenvironment, and the effect of drugs in the presence of an intact immune system [126,127,128], and can be successfully edited using the CRISPR/Cas9 system [48]. Their main limitation is that murine-based therapies in a murine system do not guarantee similar effects in humans [129,130].

Xenograft mouse models entail the transplantation of human lymphoma cells (either patient-derived, PDX, or cell lines, CDX) into immunodeficient mice (Figure 2D), thereby favoring tumor engraftment [93]. CDX models, e.g., those generated by SU-DHL-4 injection, are useful for initial drug screenings, but cell lines contain native mutations that may not be observed in the comparative human lymphoma [113,126,131,132,133]. PDXs, on the other hand, retain primary lymphoma characteristics, including TME, genetic signature and refractoriness to treatment. This allows for the rigorous testing of drug treatments and therapeutic regimens for lymphomas and is useful for personalized drug therapy [93,131,134,135]. Unfortunately, engraftment rates tend to be low, and PDXs are commonly transplanted into immune-compromised mice to avoid rejection, limiting the opportunities to study immune interactions [93,136]. This limitation can be overcome by serial engraftment, which involves passing a successfully engrafted lymphoma into other mice. However, it leads to the gradual loss of primary tumor characteristics, which is an undesirable side effect [136].

The use of immunodeficient mice populated with a human immune system (humanized mice) to study immune interactions is becoming increasingly common in the field [137,138]. These mice have the potential to bring novel cancer therapies to light that rely on a functional immune response and can be generated from a healthy donor or the patient’s own immune cells. In the B-cell lymphomas field, humanized mice are used to investigate the transition of EBV to B-cell lymphomas [139,140].

### 3.4. Back to the Bench: À La Carte-Engineered Spheroids

As previously mentioned, the interaction of malignant B-cells with stromal fibroblasts, endothelial cells, and other immune cells is key to the survival and progression of DLBCL [141,142,143]. 2D cultures lack the complexity, intercellular interactions, hypoxic conditions, and metabolic alterations found in the primary tumor [144,145]. This has led to 3D models being increasingly used in recent years [146]. These provide promising means of overcoming this limitation by recapitulating the organ structure, microenvironment, and physiological function, reliably mimicking disease states [147].

In vitro 3D models can be generated from cell lines or directly from the primary tumor (Figure 2C). However, unlike solid tumors, which can spontaneously self-organize into tissue-like structures (organoids), lymphoma organoids (also referred to as spheroids) are artificially engaged in 3D structures and enriched in the cell types present in the primary tumor [148,149].

Even though the standardization of 3D DLBCL spheroids has been challenging, the availability of different matrices reproducing specific components of the original extracellular matrix (ECM), such as polyethylene glycol-maleimide, methylcellulose or alginate, and puramatrix, has made it possible to generate DLBCL spheroids in the presence or absence of stroma cells. These maintain the cell shape, polarization, and spatial constraints seen in the physiological environment [150,151,152]. In addition, the biological responses and immune interactions between tumor and immune cells in the spheroids replicate those seen in mouse models [95,135]. However, the need to generate these spheroids afresh and the wide variety of matrices required to grow them have also complicated the use of standardized spheroids and limited the generation of highly characterized DLBCL spheroid biobanks.

Patient-derived lymphoma organoids (PDOs) reproduce the actual interactions within a patient’s own immune system to eradicate tumor cells, and recapitulate the transcriptional, mutational profile, and therapy response of the primary tumor (Figure 2B,D) [153,154]. They can be directly generated from a biopsy, preserving endogenous immune cells and other cell types (holistic approach), or can be established separately from immune cells before co-culture, which facilitates the study of specific cell–cell interactions (a simplistic approach) [155]. PDOs are a promising tool for developing and testing personalized medicine approaches [156]. However, before they can be applied as part of the daily clinical routine, organoid-based bioassays need to be standardized and improved to clinical grade.

A major limitation of organoids involving different cell types is that the co-culture conditions must be optimized to achieve an equilibrium that suits all cell types involved. Additionally, patient-derived 3D models are often created from small biopsies, in part due to the increasing use of large needle gauge punches, which may underrepresent the complexity of the whole tumor, lack immune selection, and be difficult to manipulate in vitro. Access to larger biopsies could facilitate optimizing these systems and broaden the scope of these models beyond drug screenings.

Despite all these limitations, cell-line-derived DLBCL spheroids have already been used to evaluate germinal center reactions, the tumor microenvironment and therapeutic potential and resistances [95,135,157]. Spheroid systems have been used to study the signaling and epigenetic events essential for the germinal center reaction, such as the EZH2 regulation of GC B cell proliferation through the repression of CDKN1A, which cannot be examined in mouse models because mice lacking *Ezh2* do not form germinal centers [158]. Cell-line-derived and patient-derived DLBCL organoids have also been used to test combination therapies in genetically defined backgrounds and to design anti-lymphoma regimens for ABC-DLBCL [159,160].

More complex systems, such as organ-on-chip models and microfluidics-based platforms, recapitulate blood-flow conditions and accurately reproduce the cell interactions that occur in the DLBCL tumor microenvironment (Figure 2B) [151,161]. Indeed, the combination of on-chip platforms with 3D matrices to mimic ECM composition and architecture can be used to incorporate molecular and cellular gradients such as those found in patients and the study of drug kinetics.

Overall, 3D models are very likely to facilitate faster and less expensive transferable drug development [162]. They are emerging as a better model than conventional established 2D cell lines for drug screening [163], because they more closely recapitulate the TME of primary tumors. Although animal models remain essential, 3D models could eventually reduce the use of animals for drug testing and, therefore, lower costs, shorten the time taken to obtain results, and enable the avoidance of ethical issues.

As useful as 3D culture models already are, the future of organoid systems points to the generation of 4D systems that dynamically respond to microenvironmental changes in the biochemical, physicochemical, and biophysical properties of the microenvironment, to offer more accurate physiological models to recreate lymphoid-like structures.

## 4. Conclusions

Massive sequencing has been used to identify genetic signatures in lymphomas that help predict which patients could benefit from the standard treatment and their outcome, and to enable the proposal of alternative therapeutic approaches.

Functional assays should provide the information needed to fill the knowledge gap between genomic information and treatment. Established experimental models, such as tumor cell lines and mice, do not completely capture the complex genetic and epigenetic landscape that each patient presents, so the translation of findings from these models to the clinical setting is not yet efficient. However, the emergence of 3D models such as spheroids/organoids, which can be generated from several cell types and reproduce the interactions of the patients’ immune system with their tumor, are promising tools that could be used in personalized medicine approaches in the near future (Figure 3).

We believe that combining gene editing and 3D cultures will soon allow for the replication of the genetic signatures identified in lymphoma patients, in cell lines, or primary cells in vitro, reproducing the tumor microenvironment and immune interactions. This will very likely prompt a shift in our understanding of the relevance of these clusters of mutations in therapy responses and lymphoma progression, and, more importantly, may inform the design of patient-specific therapies.

## Figures and Tables

**Figure 1 cancers-13-02893-f001:**
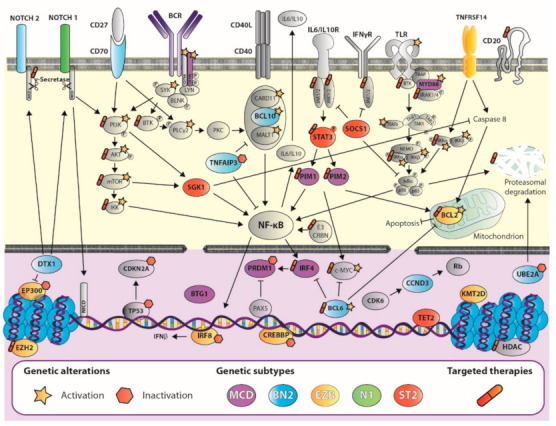
Diffuse large B-cell lymphoma complexity: untangling the networks underlying genetic subtypes. Schematic view of the most relevant genetic events in B-cell physiology pathways for each genetic subtype (GS) highlighted in purple (MCD), light blue (BN2), yellow (EZB), green (N1), or red (ST2), as described in the figure key. Mutations in CD79A and CD79B and SYK lead to chronic activation of the B-cell receptor (BCR; of special importance in the MCD GS), which triggers the activation of SYK, BTK, and PKC promoting the formation of the CARD11-BCL10-MALT1 complex. Mutations in genes that encode all the components in this complex can be identified in a subset of DLBCLs and are especially relevant for the BN2 GS. Mutations in genes that encode MYD88, IRAK1, and IRAK4 activate Toll-like receptor (TLR) signaling, which is further promoted by alterations in positive regulators downstream TLR, such as TRAF6. Alterations in Notch signaling, binding of CD27 to CD70, and BCR activation also lead to abnormal activation of PI3K-AKT-mTOR pathway, which also presents genetic alterations in this disorder. Most of these pathways converge in the activation of IKK and downstream NF-κB signaling, promoting lymphomagenesis. Tumor cell survival is boosted by alterations in the BLC2 and BCL6 axis, TP53, and imbalances in cytokine related pathways: interleukins (IL), interferon gamma (IFNγ), or tumor necrosis factor (TNF), which overall lead to increased survival and decreased apoptosis in these cells. Alterations at the nuclear level are also common in all genetic subtypes (except for the N1 subtype), and involve p53, DNA damage pathway, epigenetic regulators, and other components involved in proliferation and survival cell processes. Currently, there are targeted therapies against several components downstream these pathways (tagged with a pill icon). Activating genetic events are highlighted with a star, while those involving inactivation are flagged with a hexagon.

**Figure 2 cancers-13-02893-f002:**
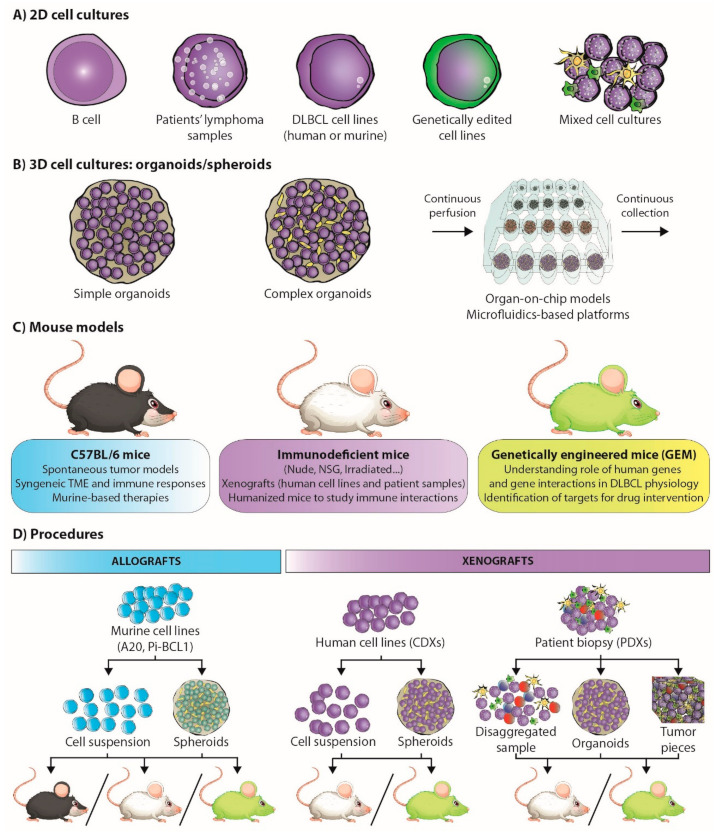
Preclinical models in lymphoma research and procedures. Research models in B-cell lymphomas include: (**A**) two-dimensional (2D) cell cultures comprising B-cells, patient lymphoma samples, DLBCL cell lines, gene edited cell lines, or mixed-cell cultures; (**B**) three-dimensional (3D) cell cultures, such as simple organoids involving a single cell type, complex organoids involving several cell types, and more complex systems, such as organ-on-chip models and microfluidics-based platforms; (**C**) mouse models, with CD57BL/6, immunodeficient, or genetically engineered mice (GEM). Their uses are summarized in the boxes below the mice. (**D**) All these elements can be combined and used to develop a variety of procedures involving cells from the same species (syngeneic allografts, left) or from different species (xenografts, right).

**Figure 3 cancers-13-02893-f003:**
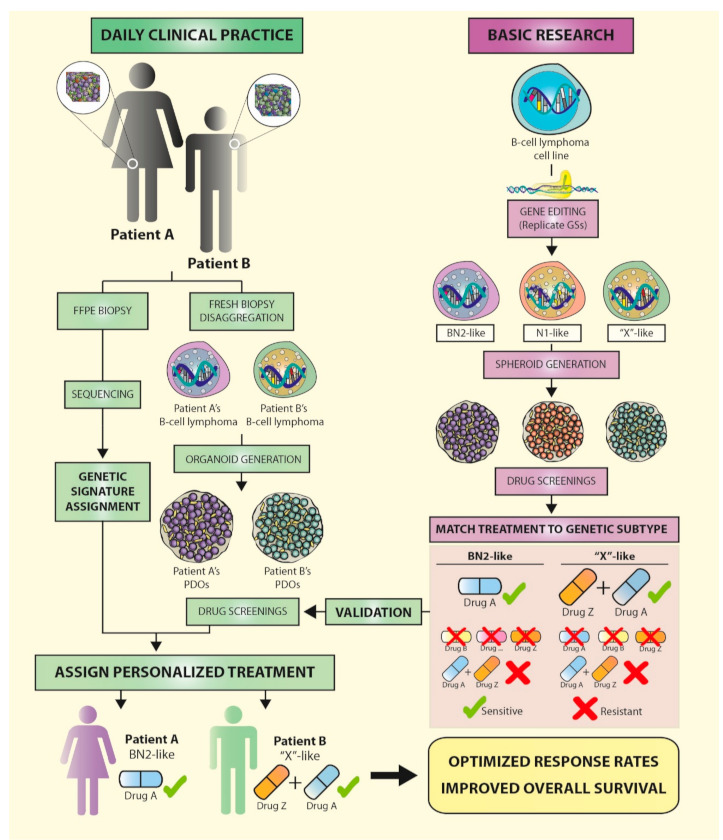
The near future for B-cell lymphoma patients: genetic signatures lead to personalized treatments. Recent studies have identified genetic signatures in diffuse large B-cell lymphoma (DLBCL) patients that can be used to predict outcome and therapy response. In the near future, a combination of newly adapted routines in daily clinical practice (left) and basic research (right) are very likely to lead to personalized treatments for B-cell lymphoma patients, based on the genetic profile of their biopsies. To this end, gene edition of DLBCL cell lines in the laboratory will enable the replication of the DLBCL genetic subtypes (GS) identified in patients (BN2, N1, EZB, MCD, and ST2). 3D organoid-based drug screenings will establish the sensitivities of the different genetic variants to single drugs or drug combinations, at the dose and posology levels, while maintaining the lymphoma microenvironment and intercellular interactions. Once a GS has been matched to a certain treatment regimen, drug sensitivity should be validated in organoids derived from fresh patient biopsies (patient derived organoids, PDOs). Inclusion of biopsy sequencing using targeted panels of selected genes as part of the daily clinical routine will eventually allow personalized treatments to be directly assigned based on the GS of patient biopsies, optimized treatment-response rates, and improved overall survival. FFPE: formalin-fixed, paraffin-embedded.

**Table 1 cancers-13-02893-t001:** DLBCL genomics: the genetic signature helps predict patient outcome. Comparison of the newly proposed genetic subtypes in DLBCL. The names of the genetic subtypes proposed by each group are highlighted in bold. COO: cell of origin; ABC: activated B-cell like; GCB: germinal center B-cell like; amp: amplification; transl: translocation.

Wright and Schmitz [28,31]	Chapuy [32]	Lacy [33]	Pedrosa [34]	COO and Patient Outcome
**MCD**	**C5**	**MYD88**	**MCD^2-S^**	ABC
*MYD88*^L265P^, *CD79B*, *PIM1*, *CDKN2A*	*MYD88*^L265P^, *CD79B*, *PIM1*, *PRMD1*, *ETV6*	*MYD88*^L265P^, *CD79B*, *PIM1*, *ETV6*, *CDKN2A*	*MYD88*, *CD79B*, *PIM1*, *PRDM1*, *PIM2*, *BTG1*	Bad prognosis
**BN2**	**C1**	**NOTCH2**	**BN2^2-S^**	ABC and GCB
*NOTCH2*, *BCL6* transl., *SPEN*, *TNFAIP3*, *CD70*, *UBE2A*, *BCL10*, *CCND3*	*NOTCH2*, *BCL6* transl., *SPEN*, *TNFAIP3*, *CD70*, *BCL10*	*NOTCH2*, *BCL10*, *TNFAIP3*, *CD70*, *SPEN*, *CCND3*	*NOTCH2*, *BCL6* transl., *TNFAIP3*, *CD70*, *BCL10*, *UBE2A*, *CCND3*, *DTX1*	Variable prognosis
**EZB**	**C3**	**BCL2**	**EZB^2-S^**	GCB
*EZH2*, *BCL2* transl., *CREBBP*, *EP300*, *IRF8*, *MEF2B*, *GNA13*, *KMT2D*, *REL* amp.	*EZH2*, *BCL2 **, *CREBBP*, *EP300*, *IRF8*, *MEF2B*, *GNA13*, *KMT2D*	*EZH2*, *BCL2*, *CREBBP*, *IRF8*, *MEF2B*, *KMT2D*	*EZH2*, *BCL2 **, *CREBBP*, *TNFRSF14*, *IRF8*, *KMT2D*, *EP300*	Generally favorable prognosis
**N1**		**NEC**	**N1^2-S^**	Mostly ABC
*NOTCH1*		*NOTCH1*, *REL* amp., *TP53*	*NOTCH1*	Bad prognosis
**A53**	**C2**	**NEC**		Mostly ABC
*TP53*, aneuploidy	*TP53*, *REL*, *CDKN2A* loss, aneuploidy	*NOTCH1*, *REL* amp., *TP53*		Bad prognosis
**ST2**	**C4**	**SOCS1/SGK1–TET2/SGK1**	**ST2^2-S^**	Mostly GCB
*SOCS1*, *TET2*, *SGK1*	*SGK1*, *HIST1H1E*, *NFKBIE*, *BRAF*, *CD83*	*SOCS1*, *TET2*, *SGK1*	*SOCS1*, *TET2*, *SGK1*, *STAT3*	Favorable prognosis

* Mutation and translocation.

## Data Availability

Not applicable.

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
