# Peer review of "An Overview on Diffuse Large B-Cell Lymphoma Models: Towards a Functional Genomics Approach"

_cancers, 2021, doi:10.3390/cancers13122893_

Round 1
Reviewer 1 Report
The authors provide a comprehensive overview of the genetic background in DLBCL, existing mouse models and the set up of 3D cultures in DLBCL. The illustrations are very nice.
Minor points:
- Paragraph on page 13, line 277 to 282 seems in the wrong place.
- Does the title cover the subject well enough? Translation of genomics of DLBCL to treatable models maybe. I think the authors should look at their aim and design a better title, so it will be more specific and more people will find it!
Author Response
Comment: Paragraph on page 13, line 277 to 282 seems in the wrong place.
Action: The paragraph has been rephrased (page 14, lines 296-300)
“The B-cell receptor (BCR) pathway, key to B-cell survival, is an especially relevant therapeutic target as its expression is restricted to B-cells, revealing potential pharmacological agents specific to these cells. Importantly, BCR signaling is chronically active in ABC-DLBCLs, with the consequent canonical NF-κB pathway activation [5,6,61].”
Comment: Does the title cover the subject well enough? Translation of genomics of DLBCL to treatable models maybe. I think the authors should look at their aim and design a better title, so it will be more specific and more people will find it!
Action: The title has been modified as suggested by the reviewer. The current title of the manuscript is “An overview on diffuse large B cell lymphoma models: towards a functional genomics approach”. We have also changed the Running title to “Functional genomics in DLBCL”.
Reviewer 2 Report
Dear Authors,
the present manuscript describes efficiently many of the critical aspects of DLBCL molecular landscape. The title is captivating and the text catches some relevant technological and scientific topics.
There are, however, some issues.
Major
- The English language needs a revision throughout the manuscript (grammar, punctuation).
- Page 6, line 143-146: the sentence is not understandable. The whole paragraph "Advances in DLBCL genomics..." should be checked again for clarity. It is clear at the beginning but becomes confused later on.
- Page 12, line 273: BCL6 in DLBCL is not always overexpressed. Literature reports differences among DLBCL-subtypes in terms of protein levels of BCL6. Please, check on the recent paper by Jeremy S Abramson "Hitting back at lymphoma...", Cancer 2019. doi: 10.1002/cncr.32145.
- A relevant layer of epigenetic regulation in lymphomas is represented by methylation/demethylation processes. This topic must be included in the manuscript since methylation profiles may affect gene expression and tumor behaviour. Please, consider these recent references:
- Alberto J Arribas et al. Blood 2015 doi: 10.1182/blood-2014-08-596247.
- Frazzi R. et al. Leukemia Research 2017 DOI: 10.1016/j.leukres.2017.02.012.
- Pengfei Liu et al. Mol Med Rep 2017. DOI: 10.3892/mmr.2017.7058
Minor
- page 3, line 72: change "DLBLC" with DLBCL and use the square brackets in addition to the round ones.
- page 3 line 78: please change the sequence "..if we are eventually to be able to provide better treatment.."
- page 4, line 93: homogenize the use of singular/plural
- Please, check the correct use of the abbreviation DLBCL throughout the manuscript.
Author Response
Major Comment: The English language needs a revision throughout the manuscript (grammar, punctuation).
Action: The English language has been revised using the editing services of the journal as suggested by the reviewer.
Major Comment: Page 6, line 143-146: the sentence is not understandable. The whole paragraph "Advances in DLBCL genomics..." should be checked again for clarity. It is clear at the beginning but becomes confused later on.
Action: The paragraph has been rephrased to facilitate its understanding.
Major Comment: Page 12, line 273: BCL6 in DLBCL is not always overexpressed. Literature reports differences among DLBCL-subtypes in terms of protein levels of BCL6. Please, check on the recent paper by Jeremy S Abramson "Hitting back at lymphoma...", Cancer 2019. doi: 10.1002/cncr.32145.
Action: We have moderated the affirmation on page 16, line 361, to “Additionally, BCL6 is overexpressed in over half of DLBCL cases, regardless the COO.”.
Major Comment: A relevant layer of epigenetic regulation in lymphomas is represented by methylation/demethylation processes. This topic must be included in the manuscript since methylation profiles may affect gene expression and tumor behaviour.
Action: We agree with the reviewer’s view and have added a paragraph addressing this issue on page 5, lines 99-103.
“Epigenetic processes, such as methylation and demethylation, are responsible for the dynamic plasticity of B-cells and affect tumor behavior (for a recent review see [1]). Interestingly, many of the genetic mutations that occur in DLBCL affect genes involved in chromatin remodeling (EZH2, CREBBP, KMT2D, etc.) and other epigenetics functions such as DNA methylation (TET2).”
Minor Comment: page 3, line 72: change "DLBLC" with DLBCL and use the square brackets in addition to the round ones.
Action: The paragraph has been rephrased as follows (page 5, lines 92-96):
“Indeed, the proposal of alternative treatments for diffuse large B-cell lymphoma (DLBCL) patients who do not respond to standard immunotherapeutic regimens (up to 30-40%) is mostly empirical, based on trial and error, and does not consider their genetic complexity.”
Minor Comment: page 3 line 78: please change the sequence "..if we are eventually to be able to provide better treatment.."
Action: The sentence has been rephrased as suggested by the reviewer (page 5, lines 107-108):
“We need to expand this concept beyond genomics to eventually provide better treatment options for patients in need of alternative therapeutic approaches.”
Minor Comment: page 4, line 93: homogenize the use of singular/plural
Action: We have homogenized the use of singular in the sentence following the reviewer’s recommendation (page 6, line 128):
“Two subtypes are recognized based on the gene expression profile:”
Minor Comment: Please, check the correct use of the abbreviation DLBCL throughout the manuscript.
Action: DLBCL spelling has been revised throughout the manuscript and changes have been made in page 5, line 94, and page 6, line 141.
Reviewer 3 Report
It is an impressive review on the findings of genomics in DLBCL by various technical approaches. The different methods have been reviewed with their limitations. Some are now classical some more experimental. The idea was to have markers of pathways of deregulation of lymphomas cells to elaborate targeted treatments. With the evolution of technologies, new genetic signatures help to predict outcome.
In fact, if these data are explaining in part the biology of the disease, in vitro targeted treatment in DLBCL does not guarantee the effect in human. It has been disappointing to see that BTK inhibitors have limited effect in DLBCL at least less than in other subtypes of lymphoma. The same comments can be made for NFK B and many other agents.( page 10-11)
The article should better reflect these findings.
One reason is the complexity of multiple pathways, but it does not explain why 70% of DLBCL are cured with rough immunochemotherapy.
Figure 2 and 3 are nice and explain the different layers of findings but they are hypothesis generating with only partial success right now in DLBCL.
The main new input in treatment in R/R DLBCL was CAR T and the tumor micro environment. I hope that fascinating bio Patients-derived organoids, PDOs, will change the understanding of the lymphoma biology. However, it seems that the system remains within the limit of cellular approach and restricted to test new drugs. Can you explain more this technology ( time, cost, material) behind to motivate clinical researchers. . Large needle gauge biopsies are now widely admitted. Can you use the material provided.?
Author Response
Comment: In fact, if these data are explaining in part the biology of the disease, in vitro targeted treatment in DLBCL does not guarantee the effect in human. It has been disappointing to see that BTK inhibitors have limited effect in DLBCL at least less than in other subtypes of lymphoma. The same comments can be made for NFK B and many other agents.( page 10-11). The article should better reflect these findings.
Action: We have included a series of sentences addressing this issue:
Page 14, line 309: “However, clinical trials have shown that BTK inhibitors have a limited effect on DLBCL compared to other lymphoma subtypes [64].”
Page 15, line 323-325: “Despite the increasing efforts in the discovery and generation of drugs targeting NF-κB, the clinical trials performed to date have limited applicability for DLBCL patients [68,69].”
Comment: The main new input in treatment in R/R DLBCL was CAR T and the tumor micro environment. I hope that fascinating bio Patients-derived organoids, PDOs, will change the understanding of the lymphoma biology. However, it seems that the system remains within the limit of cellular approach and restricted to test new drugs. Can you explain more this technology ( time, cost, material) behind to motivate clinical researchers. . Large needle gauge biopsies are now widely admitted. Can you use the material provided.?
Action: We have added a comment on this on page 23, lines 518-523:
“Additionally, patient-derived 3D models are often created from small biopsies, in part due to the increasing use of large needle gauge punches, which may underrepresent the complexity of the whole tumor, lack immune selection, and be difficult to manipulate in vitro. Access to larger biopsies could facilitate optimizing these systems and broaden the scope of these models beyond drug screenings.”